# Optimization of Collagen Scaffold with Cultured Autologous Chondrocytes for Osteochondritis Dissecans of the Knee: A Case Report

**DOI:** 10.3390/reports7030062

**Published:** 2024-07-30

**Authors:** Nicolas Valladares, Gibran J. Jacobo-Jimenez, Nathaniel Lara-Palazuelos, Maria G. Zavala-Cerna

**Affiliations:** 1Sports Medicine Division, Rehabilitem, Guadalajara 44100, Mexico; nvalladares@hotmail.com (N.V.); gibranjjacobo@gmail.com (G.J.J.-J.); 2Unidad Académica Ciencias de la Salud, Universidad Autónoma de Guadalajara, Zapopan 45129, Mexico; nathaniel.lara@edu.uag.mx

**Keywords:** matrix-induced autologous chondrocyte implantation, cartilage defect, clinical outcomes, osteochondritis dissecans, case report

## Abstract

The treatment of osteochondritis dissecans of the knee has always been a challenge for orthopedic surgeons. We present a case report of a 38-year-old male with severe right knee pain after suffering from an indirect trauma and axial rotation of the knee, limiting knee functionality and impeding his ability to walk, with a diagnosis of osteochondritis dissecans in the trochlea of the knee, who underwent arthroscopic treatment with matrix-induced autologous chondrocyte implantation (MACI). After the surgery, a physical therapy protocol for MACI was implemented, and magnetic resonance images with cartilage mapping were used to evaluate the recovery of the lesion. A total recovery was observed and evaluated with the modified Cincinnati knee rating system (mCKRS). A discussion is provided with evidence and general recommendations for the use of MACI in the treatment of adult OCD of the knee as a possible alternative to conventional treatments. Our case shows a rapid improvement in pain and functionality 2 months after surgery that progressed to full recovery within 6 months.

## 1. Introduction

Osteochondritis dissecans (OCD) is an idiopathic, focal, subchondral bone abnormality that can cause instability or detachment of a bone fragment and overlying articular cartilage, with subsequent progression to osteoarthritis [1]. OCD is usually regarded as either juvenile OCD (occurring with an open epiphyseal plate) or adult OCD (after the physis has closed) [2]. The overall incidence rates in adults per 100,000 person-years are 3.42 for all OCD cases and 1.21 for knee OCD [3]. Juvenile OCD is more common in those aged 6–19 years, with an incidence of 9.5 per 100,000, with an increased risk in those who are >12 years [4].

The classic site of knee OCD is the posterior-lateral aspect of the medial condyle (63.6%), compared with the inferior-central aspect of the lateral condyle (32.5%), the inferomedial aspect of the patella (1.5% to 10%), and the trochlea (less than 1%) [4,5]. The etiology of OCD is not completely understood. A recent systematic review reported that OCD can be caused by biological and/or mechanical factors, within the first group of factors we can include genetic, inflammatory, ossification-deficit, and poor vascular supply have been described. The mechanical etiological factors include the hypotheses of injury and overuse, tibial spine impingement, presence of discoid meniscus, and biomechanical alteration [6].

The symptoms depend on the lesion location and stage. Stable lesions can cause nonspecific symptoms, including vague or intermittent pain, whereas unstable lesions or loose bodies can cause mechanical symptoms, including catching or locking. Restricted range of motion and joint effusion may signify unstable lesions or loose bodies [7].

Radiography is the first step. An anteroposterior view, a lateral view, and a tunnel view with the knee flexed at 60 °C should be obtained. A skyline view is required if an OCD lesion of the patella or trochlea is suspected [1]. MRI is the method of choice as the second step in an imaging workup [2]. T1-weighted sequences allow lesion size measurement, while T2-weighted sequences with cartilage mapping provide information on articular cartilage integrity and reactive marrow edema adjacent to the affected subchondral bone [7].

The operative treatments for cartilage defects can be divided into palliative, reparative, restorative, and reconstructive procedures. Palliative procedures include lesion debridement and loose body removal, with the single goal of alleviating mechanical symptoms. Reparative procedures include microfracture to achieve bone marrow stimulation. Reconstructive procedures include mosaicplasty or osteochondral autograft transplantation (OAT) and matrix-induced autologous chondrocyte implantation (MACI), where the purpose of the treatment is to fill the articular defect with hyaline-like cartilage tissue [8].

The treatment for adult knee OCD with lesions classified as stage III requires a treatment to fill the articular defect; however, not all approaches have reported favorable outcomes. Microfractures, however, have shown a low repair rate due to the resultant unstable fibrocartilage, which usually degenerates within less than 5 years [9,10]. Mosaicplasty, which consists in transplanting a healthy osteochondral area to another affected area, is suitable in lesions smaller than 2 cm^2^; however, it can place load-bearing cartilage in places where friction predominates, leaving an area without cartilage, which could potentially lead to the generation of fibrosis [11]. MACI is considered a third-generation technique for cartilage repair that consists of two stages: an initial arthroscopy is carried out to perform a cartilage biopsy, followed by open or arthroscopic surgery to implant the cultured chondrocytes into the cartilage defect. Even though clinical benefits associated with increased survival of the graft, patient satisfaction, and faster return to daily activities have been documented with the use of MACI, there is a lack of consensus related to its status as the treatment of choice for a specific surgical technique or timing of intervention, which can translate into an absence of standardized treatment. Therefore, our purpose is to share our experience with the use of MACI, hoping to contribute to a full standardization of this treatment to improve clinical outcomes in a significant number of patients affected by this condition.

## 2. Case Report

The present case is presented according to CARE guidelines [12]. A 38-year-old male presented to our clinic with severe right knee pain after suffering from an indirect trauma and axial rotation of the knee. The pain limited knee functionality, impeding the patient from walking for the previous two days. Upon evaluation, the patient presented an antalgic gait at the expense of the right knee with pain in the patellofemoral region and in the medial intra-articular line. The knee exploration revealed ranges of motion (ROM) of 40° of knee flexion and −3° of extension due to pain; crepitus was noted during the ROM of the knee. The ballottement test was positive for knee effusion; the medial McMurray and Appley tests were positive; and the Lachmann anterior and posterior tests were negative (Figure 1). To evaluate the patient’s functional status, we used the modified Cincinnati knee rating system (mCKRS), and the score was 24. This system consists of 12 scored questions that cover the domains of pain, swelling, function, and activity level, with 100 representing excellent knee function and 0 representing poor knee function. Overall, the mCKRS can be interpreted as follows: <30 poor, 30–54 fair, 55–79 good, and >80 excellent [13].

A knee MRI was ordered to identify the type and extent of the lesion. Findings revealed a medial meniscal posterior horn tear without evidence of osteochondral lesion; nevertheless, in a postoperative inspection of the images, we found changes in the trochlea of the knee suggestive of osteochondral lesion (Figure 2). The preoperative diagnosis was medial meniscal posterior horn tear of the right knee.

An arthroscopic surgical intervention was performed on the right knee of the patient for reparation of the medial meniscal posterior horn tear. During the procedure, a chondral lesion was identified at the level of the trochlea measuring 25 mm × 15 mm, involving the cartilage in full depth with exposure of the subchondral bone (Outerbridge grade IV); additionally, a free chondral body of similar dimensions was identified (Figure 3).

A trans operative diagnosis of osteochondritis dissecans of the knee grade IV in the Diapola Classification [14] was established. The chondral body was removed and sent for histopathologic study; afterwards, the meniscus repair was conducted with the use of all-in meniscal sutures. Before closure of the arthroscopic portals, a joint drainage (Drenovac) was placed, and wounds were closed. The patient was discharged the same day of the surgery with etoricoxib 120 mg once daily and levofloxacin 750 mg once daily for 7 days to prevent wound infections.

After considering the extent of the chondral lesion (>2 cm^2^), the young age of the patient, and the evidence related to the long-term benefits of the use of MACI, we offered this type of intervention for chondral repair treatment; however, before obtaining a biopsy to be used for chondrocyte culture, insurance approval was necessary.

The patient started physical therapy for pain and inflammation management, as well as for the meniscal suture protocol, which consists in progressive mobilization with partial progressive weight-bearing movement. The patient completed the meniscal repair protocol, an approval was obtained from the insurance company, and a second arthroscopic procedure with the aim of obtaining a cartilage biopsy for chondrocyte culture was scheduled. A small cartilage sample was taken from the non-weight-bearing area of the trochlea using an arthroscopic chondrotome and then transferred into a serum-free transport medium (chondrocyte medium I) and sent to Gencell Biotechnology in Mexico City for culturing and processing (Figure 4).

Chondrocytes were isolated by enzyme digestion and resuspended into chondrocyte medium I with 10% bovine fetal serum. Cells were transferred several times until a confluence of 100% was reached with a cell count of 5 million per cm^2^ (Figure 5).

Cells were transferred (3–5 million) into a collagen I/III matrix (Chondro-Gide^®^ Wolhusen, Switzerland). The matrix was packed, sealed, and sent to be placed in the patient within the next 24 h. Three weeks after the first intervention, the patient was readmitted to apply the matrix with the cultured autologous chondrocytes. An arthroscopy surgery was performed in a tourniquet-controlled bloodless field; intra-articular arthroscopic retractors were placed to expose the defect; the chondral lesion was excised with a chondrotome; and the subchondral bone was prepped until it had a flat and uniform layer of bleeding bone with a ring of normal surrounding cartilage. The MACI (chondroMATRIX^®^, Queretaro, Mexico) was cut to size to allow for correct membrane integration, then fixed with fibrin glue (Tissucol) on top, and finally, the prominent edges were trimmed (Figure 6).

Towards the end of the surgery, the knee was flexed and extended to confirm the stability of the implant. After the surgery, the patient was admitted for physical therapy according to the rehabilitation guidelines for third-generation MACI [15], during which the patient’s knee was immobilized with a brace for 4 weeks in order to avoid shearing forces on the knee joint. Meanwhile, physical therapy was started with a protocol specific for MACI, consisting in continuous passive motion, management of pain and swelling via physical media, and the use of electrostimulation to maintain muscular tone, in addition to isometric contraction of the quadriceps, hamstrings, adductors, and gluteal musculature [16]. At 6 weeks after the surgery, a complete ROM was achieved, and mobility of the knee was completed. Lower limb strengthening and gait re-education were completed 2 weeks later. Upon completing all the objectives, the patient was discharged from physical therapy.

Follow-up examinations were performed at 3 and 6 months after the surgery. Clinical improvement was evident at 3 months after the surgery with progressive movement and less pain in the right knee. At this time, the mCKRS was 88. At 6 months after the surgery, the patient presented with significant clinical improvement, with absence of pain, complete ROM, and functionality with an mCKRS of 100. At this time, a 3-Tesla control MRI of the right knee was ordered, which showed the complete regeneration of the chondral defect without a trace of injury or thinning of the cartilage compared to healthy cartilage (Figure 7). At this time, the patient was able to perform daily activities and sports without limitations or pain.

## 3. Discussion

In the present case, the diagnosis of OCD was suspected during the initial evaluation and confirmed during the first arthroscopic intervention.

Many surgical options have been proposed for the treatment of stage III and IV OCD lesions. However, there is an ongoing debate related to long-term results when different techniques are compared. Additionally, some factors have been previously associated with poor outcomes, including incongruence of the fragment or too many fragmented loose bodies [17].

Autologous chondrocyte implantation (ACI) has been reported as a good alternative to treat large lesions with decent performance over the use of mosaicplasty in the past [11,18]. However, some problems became apparent with this technique, such as the treatment of the subchondral bone, hypertrophy of the periosteal flap, and dedifferentiation of chondrocytes. This gave rise to additional alternatives, such as the matrix-supported transplantation of autologous chondrocytes (MACI).

Since the matrix supplies the natural environment of chondrocytes, it will provide an environment with adequate physical and biochemical conditions, in addition to providing a parallel columnar arrangement like that of the natural articular cartilage [19]. It has been observed that the implementation of MACI as a technique for the repair of chondral defects has significantly improved quality of life in patients, with high satisfaction rates as well as graft survival exceeding 10 years [20], and for the treatment of OCD up to 36 months, with better results compared to osteoarticular transfer systems (OATs) [21].

Compared to other techniques used to repair chondral lesions, there is a significant improvement in the quality of life of people who have undergone a MACI procedure.

Several studies have been conducted in the past to analyze MACI performance in comparison to other techniques for cartilage repair. This superiority has not been shown in a very clear form, mostly due to a lack of consensus regarding the evaluation of postoperative clinical outcomes with respect to a minimal clinically important difference. For this purpose, a meta-analysis was conducted to determine the proportion of available cartilage repair studies that either meet or exceed the minimal clinically important difference (MCID), which is a measurement of whether changes in patient-reported outcome scores reflect meaningful improvement for the patient [22]. After conducting this meta-analysis, it was evident in terms of the International Knee Documentation Committee (IKDC) and the visual analog scale for pain (VAS) used to report MCID that the microfracture technique met the MCID values for all outcome scores at short- and midterm follow-up, except for pain at midterm, while osteoarticular transfer systems (OATs) met the MCID values for all outcome scores at short-, mid-, and long-term follow-up (except for VAS at the long term) and for MACI MCID values at all times, with extended maintenance of clinical benefits. The benefit observed, especially in the long-term follow-up and beyond, is what started the dialogue to consider MACI as a superior alternative procedure. However, the studies included in this meta-analysis might be heterogenic with regards to patient characteristics and may have biased the results. Another important limitation of this study is that microfractures are typically used to treat smaller cartilage lesions [23], and by not including the size of the lesion in this analysis, there might be selection bias influencing the results in favor of this procedure, although it was not the best approach.

In this regard, a study compared the long-term follow-up of patients who underwent cartilage repair after failure of conservative treatment. The patients were treated with either MACI or mosaicplasty. The patients were followed for a 12-year period; at the end of the study, both interventions presented with satisfactory clinical and MRI evaluations; however, for larger lesions (>2 cm^2^), MACI presented with fewer failures and therefore a lower risk of reoperation compared to mosaicplasty [24]. Although the treatment groups in this study were similar in terms of demographics and cartilage lesion features, the decision for treatment was not randomized, and therefore, this study might be subject to selection bias.

A randomized controlled trial, with the purpose of examining the clinical efficacy and safety at midterm follow-up (5 years) with MACI in comparison to microfractures for patients with symptomatic cartilage defects of the knee, found superior improvements with MACI. These improvements were consistent in the knee injury and osteoarthritis outcome score (KOOS) pain and function domains after surgery and were maintained over a 5-year follow-up. Superiority was also observed in function, with improvements in daily living and quality of life at 2 years of follow-up. The MRI evaluation showed similar improvement related to defect filling for both treatment groups [25].

With respect to the treatment of unstable OCD in adults, a recent literature review performed a qualitative analysis of different treatment options and included information from follow-ups, which ranged from 2 to 17 years. The internal fixation surgical technique had acceptable rates of radiographic union and patient-reported outcome measures in skeletally immature patients. For adults with large lesions, MACI and osteochondral autograft transplantation (OAT) were found to have the best performance. However, in comparison to MACI, OAT had a higher conversion to arthroplasty [26].

A recent long-term prospective study followed 87 patients with OCD in the tibiofemoral and patellofemoral regions who were treated with MACI. Over a 10-year follow-up, the study found increased graft survival compared to microfracture. Treatment with MACI was also associated with high patient satisfaction rates [20].

To be noted, one of the common criticisms of cell-based repair techniques is their propensity for graft hypertrophy, which can lead to mechanical symptoms, pain, and the need for reoperation. However, a network meta-analysis provided conclusive evidence (with a high degree of statistical certainty) that both second-generation ACI and MACI significantly reduce the risk of graft hypertrophy in comparison with OAT and microfractures [27].

The transplantation of cultured autologous chondrocytes in a collagen scaffold is a technique that allows for the reproduction of autologous cartilage in vitro to regenerate an extensive, full-thickness chondral lesion, avoiding degeneration and the requirement for prostheses in the medium-term follow-up. Derived from variations in techniques and clinical characteristics of patients intervened with MACI, some previous studies have reported unsuccessful cases among their cohorts, which has allowed for the identification of risk factors associated with poor outcomes, especially in the short-term follow-up, including a long period of pain, the presence of a functional disorder, and larger defects (>6 cm^2^) [28]. Even though our patient had a recent previous surgery, it was for the repair of the meniscal tear and not related to the chondral lesion; a long period of pain and functional disorder, as well as a larger defect (>6 cm^2^), were not present. The absence of risk factors, in this case, might have been associated with a good clinical response.

The MACI procedure is indicated for isolated chondral and osteochondral symptomatic defects between 2 and 10 cm^2^ or with affection of the cartilage full thickness with no significant osteoarthritis [29]. Nevertheless, several drawbacks should be considered when compared to other chondral repair techniques. The first is the need for two surgical procedures, one for taking the cartilage sample and the second to apply the graft [30]; while both interventions can be performed arthroscopically, they demand advanced surgical skills to manipulate the membrane and ensure a precise adaptation to the arthroscopic defect, thereby achieving an optimal fit for implantation. As with any surgical intervention, the potential for complications during each procedure may outweigh the benefits. Several attempts have been made to avoid the first step; some include the use of human-induced pluripotent stem cells (hiPSCs) obtained from the patients’ skin or peripheral blood, followed by ex vivo differentiation into chondrocytes [31]. However, there is a lack of standardized protocols for chondrocyte differentiation, and the accessibility to this technology might be limited. Hopefully, this could be overcome soon.

A second drawback is the time required for chondrocyte culture growth, estimated from 4 to 8 weeks to achieve a cell count of 5 million per cm^2^ [32]. The third disadvantage is the cost of the procedure, estimated to be up to three times higher than a scaffold alone or mosaicplasty. In addition, a lengthy postoperative rehabilitation period might be required (from 6 up to 18 months) to achieve full maturation of the graft [33]. This extended recovery time can be prolonged in patients non-compliant with the rehabilitation protocol, underscoring the importance of strict monitoring and control during the rehabilitation sessions [19].

In spite of the mentioned drawbacks, it has been demonstrated that, especially for the treatment of medium-sized lesions (>4 cm^2^), MACI provides a more cost-effective approach in comparison to microfractures [34], especially when analyzing the quality of the resulting cartilage in long-term follow-up evaluations, which might be an additional consideration for young patients.

## 4. Conclusions

The treatment of OCD with MACI is a possible and desirable alternative to conventional treatments. Our case supports rapid improvement in pain and functionality, which continues from two months after the surgery until the mid-period follow-up (6 months). The best results with this intervention are associated with small- and medium-sized chondral defects (4–6 cm^2^), as well as a short duration of symptoms and no previous surgical interventions.

## Figures and Tables

**Figure 1 reports-07-00062-f001:**
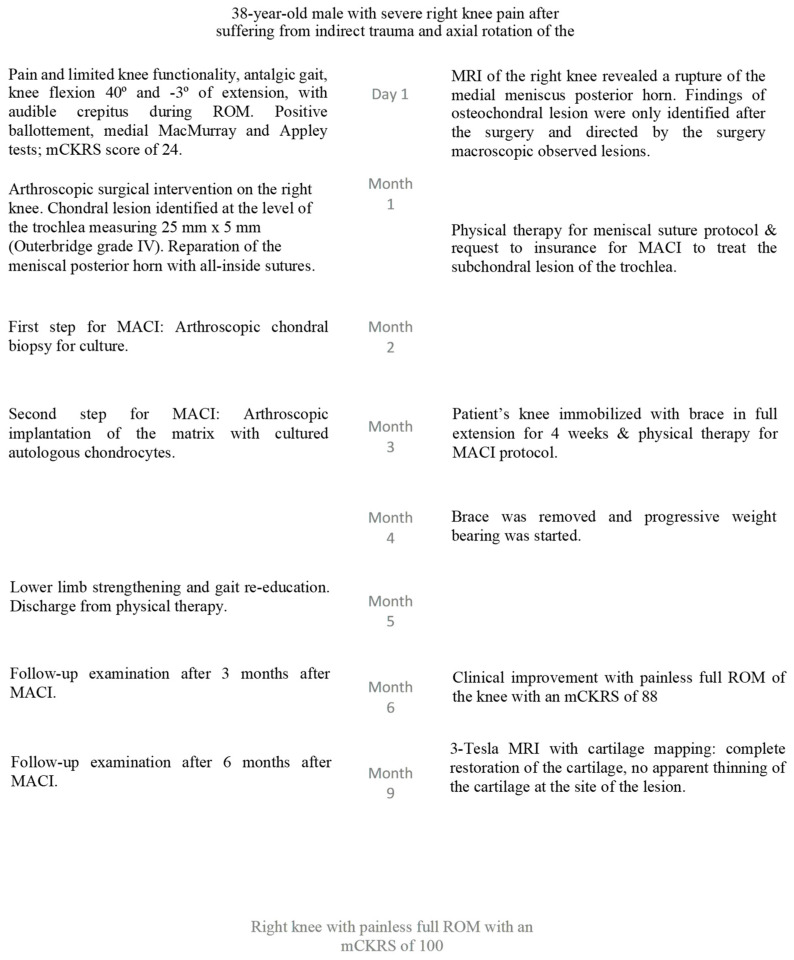
Patient timeline. Green boxes describe clinical findings, light yellow boxes represent imaging findings, dark yellow boxes describe the interventions performed on the patient, and the red box describes the outcome after 12 months of follow-up.

**Figure 2 reports-07-00062-f002:**
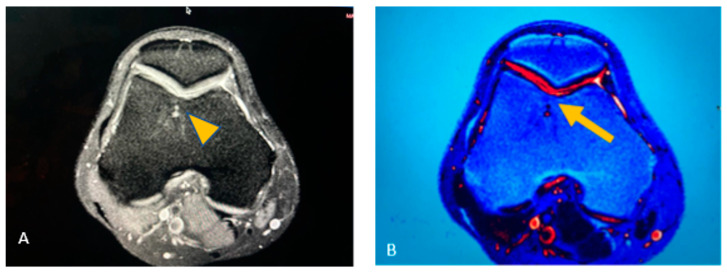
Initial knee MRI of the patient. (**A**) Axial T2 images of the MRI of the right knee, in which a subchondral lesion is noted on a second look after the surgery, directed by the surgical macroscopic observed lesions; a subchondral lesion is noted at the trochlea (arrow tip); (**B**) digital T2 map filter showing the appearance of a disruption of the cartilage (arrow).

**Figure 3 reports-07-00062-f003:**
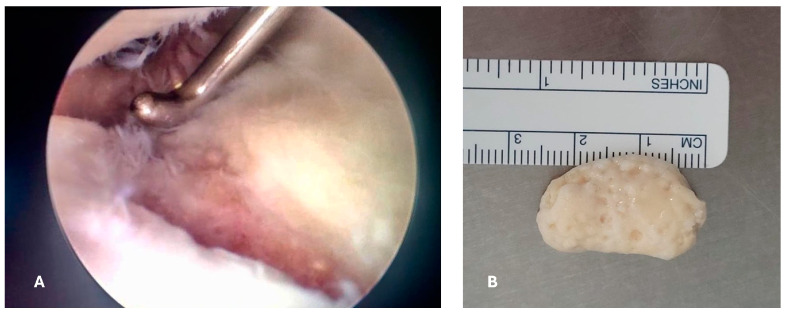
First surgical intervention. (**A**) Arthroscopic vision where the osteochondral lesion on the trochlea is visible, affecting both cartilage and subchondral bone; (**B**) macroscopic image of the free chondral body, which has the same dimensions as the osteochondral lesion on the femoral trochlea.

**Figure 4 reports-07-00062-f004:**
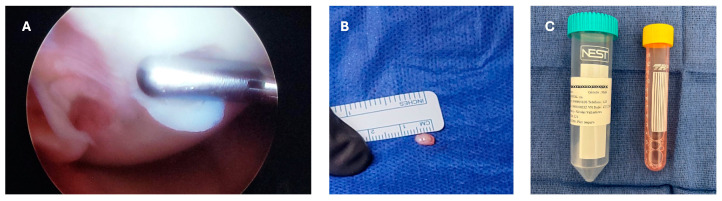
Second surgical intervention. (**A**) Arthroscopic vision of the size of the sample taken for chondrocyte culture (arthroscopic probe for reference); (**B**) macroscopic vision of the sample taken for chondrocyte culture; (**C**) sample transferred into a tube with serum-free transport medium to be sent for chondrocyte culture.

**Figure 5 reports-07-00062-f005:**
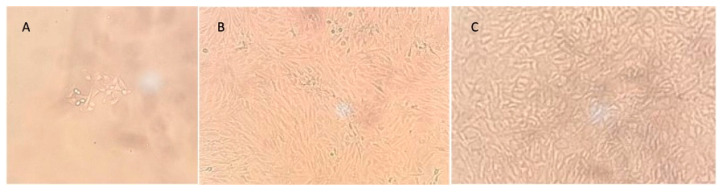
Evolution of chondrocyte culture. (**A**) Chondrocyte culture 3 days after enzymatic digestion. (**B**) Second passage of the cultured chondrocytes, with the addition of fresh cultured media enriched with 10% fetal bovine serum. (**C**) After several days of condensation of the second passage, chondrocytes became packed, acquired a stacked appearance, and could be added to the scaffold. Cell viability was routinely tested. Photographs were taken with an Optika IM-3 inverted microscope (10×), observed in bright field without staining.

**Figure 6 reports-07-00062-f006:**
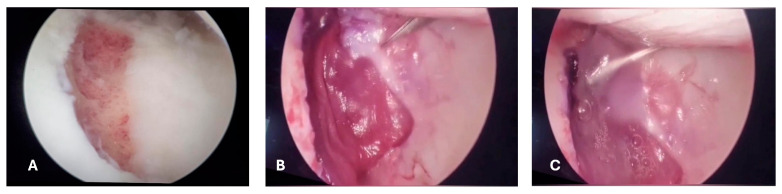
(**A**) Arthroscopic vision of the femoral trochlea with debridement of the subchondral bone. (**B**) Arthroscopic application of the matrix to fill the femoral defect. (**C**) Matrix in place with addition of a fibrin gel (Tissucol) for implant stabilization.

**Figure 7 reports-07-00062-f007:**
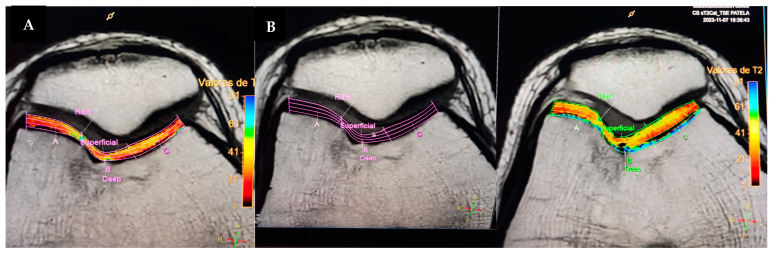
A 3-Tesla control MRI of the right knee 6 months after the surgical procedure. (**A**) T1 axial image with measurement of the cartilage in the site of MACI, with normal appearance and length. (**B**) Cartilage measurements at the site of the implant in comparison with the rest of the trochlea, with consistency throughout.

## Data Availability

The original contributions presented in the study are included in the article, further inquiries can be directed to the corresponding author.

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
