# Peer review of "Optimization of Collagen Scaffold with Cultured Autologous Chondrocytes for Osteochondritis Dissecans of the Knee: A Case Report"

_reports, 2024, doi:10.3390/reports7030062_

Round 1
Reviewer 1 Report
Comments and Suggestions for Authors
This article reports on the application of the autologous chondrocyte implantation technique in a 38-year-old male patient. The overall structure of the article is coherent, the data is sufficient, and the conclusions are valid. However, there are still some questions that need to be addressed by the authors.
1.The patient was restricted from activities for 4 weeks after the transplant, but the article does not cite any references or provide a detailed explanation for choosing this 4-week period.
2.The authors did not discuss how to optimize the time between harvesting autologous cartilage and preparing the cartilage scaffold. The longer this period, the less favorable it is for the patient.
3.The authors should discuss the drawbacks of this technique in detail and propose potential solutions to address these drawbacks.
4.The following articles may provide useful references for this paper: Biomaterials Translational, 2023, 4(1): 18. Biomaterials Translational, 2024, 5(1): 89. Cell Reports Medicine, 2023, 4(8).
Author Response
1.The patient was restricted from activities for 4 weeks after the transplant, but the article does not cite any references or provide a detailed explanation for choosing this 4-week period.
Response: We appreciate the comment, we have added references and an explanation for the 4-week period restriction from activities.
2.The authors did not discuss how to optimize the time between harvesting autologous cartilage and preparing the cartilage scaffold. The longer this period, the less favorable it is for the patient.
Response: We have added an explanation of the time it takes for the culture development, as well as, experimental available options.
3.The authors should discuss the drawbacks of this technique in detail and propose potential solutions to address these drawbacks.
Response: Thank you for your comment, we have added drawbacks of this technique into the discussion section.
4.The following articles may provide useful references for this paper: Biomaterials Translational, 2023, 4(1): 18. Biomaterials Translational, 2024, 5(1): 89. Cell Reports Medicine, 2023, 4(8).
Response: Thank you for your suggestion, however, they refer to different pathologies and it was difficult to include them in our case.
Reviewer 2 Report
Comments and Suggestions for Authors
Dear Author's
Thank you for the opportunity to read the results of Your article. I suggest minor corrections, namely; - in the abstract, please add 2-3 summary sentences - i.e. what was the final result of the treatment - I suggest Case report instead of Case presentation - Patient started physical therapy - i.e.? (please describe briefly) - the text of the article include the MACI protocol - please describe it, not everyone knows what it isbest regards
Author Response
Thank you for the opportunity to read the results of Your article. I suggest minor corrections, namely; - in the abstract, please add 2-3 summary sentences - i.e. what was the final result of the treatment.
Response: Thank you, we have added this information into the abstract.
I suggest Case report instead of Case presentation –
Thank you for your suggestion, we have changed the subtitle to Case presentation.
Patient started physical therapy - i.e.? (please describe briefly) - the text of the article include the MACI protocol - please describe it, not everyone knows what it is
Thank you for your suggestion, we have added a brief description of the MACI protocol, as well as added citation for the protocol.
Reviewer 3 Report
Comments and Suggestions for Authors
1. Please report the case adhering to the CARE guidelines.
2. Submit the filled-in CARE checklist.
3. Can quality of life assessment be carried out?
4. Can the treating physician provide a global improvement scale before and after the intervention?
Author Response
- Please report the case adhering to the CARE guidelines.
Response: Thank you for your observations. We have reported the adhesion to CARE guidelines,
- Submit the filled-in CARE checklist.
Response: We have added this as a supplementary file.
- Can quality of life assessment be carried out?
Response: Quality of life assessment was not performed.
- Can the treating physician provide a global improvement scale before and after the intervention?
Response: We did include the modified Cincinnati Knee Rating System (mCKRS) in figure 1. We have added a brief explanation about the rating system as well as the measurement before the intervention, since previously it was only mentioned in Fig. 1.
Round 2
Reviewer 1 Report
Comments and Suggestions for Authors
Through the authors' efforts, most questions have been addressed. The overall significance of the article is well established, with ample evidence provided. In future clinical applications, the use of materials for cartilage repair or osteoarthritis holds significant potential, which can be discussed in the context of this prospect. Some of the literature we provide is related to this topic.
Author Response
Through the authors' efforts, most questions have been addressed. The overall significance of the article is well established, with ample evidence provided. In future clinical applications, the use of materials for cartilage repair or osteoarthritis holds significant potential, which can be discussed in the context of this prospect. Some of the literature we provide is related to this topic.
Response: The references suggested to review were Biomaterials Translational, 2023, 4(1): 18. Biomaterials Translational, 2024, 5(1): 89. Cell Reports Medicine, 2023, 4(8). However, we searched for them and they do not refer to OCD or the treatment that we discussed in our case report. The first article refers to cartilage tissue engineering assisted by in silico models, for reconstruction of facial cartilage. The in silico model is used to replace in vivo/in vitro experiments, which is not our case. The second article is focused on osteoarthritis and the hypoxia-inducible factor-1 alpha. Not sure how this could be included in our case, since both pathologies have different etiologic factors with osteoarthritis having an important inflammatory component. The third reference, describes a general approach to produce bone and cartilaginous structures utilizing the self-regenerative capacity of the intercostal rib space to treat a deformed metacarpophalangeal joint. Since we are only describing the regeneration of cartilage and not a complete joint, this article seems to be beyond and out of scope from the current case report.
Round 3
Reviewer 1 Report
Comments and Suggestions for Authors
The quality of the article, after revisions, now meets the standards for publication.